

# Observations of aerosol optical properties at a coastal site in Hong Kong, South China

Jiaping. Wang[1,2,3,4,*], Aki. Virkkula[1,2,3,5,6], Yuan. Gao[4,7], Shuncheng. Lee[4,*], Yicheng. Shen[1,2,3], Xuguang. Chi[1,2,3], Wei. Nie[1,2,3], Qiang.Liu[1,2,3], Zheng. Xu[1,2,3], Xin. Huang[1,2,3], Tao. Wang[4], Long.Cui[4], Aijun. Ding[1,2,3]

[1] Joint International Research Laboratory of Atmospheric and Earth System Sciences, Nanjing, China
[2] Institute for Climate and Global Change Research & School of Atmospheric Sciences, Nanjing University, Nanjing, 210023, China
[3] Collaborative Innovation Center of Climate Change, Jiangsu Province, China
[4] Department of Civil and Environmental Engineering, The Hong Kong Polytechnic University, Hung Hom, Kowloon, Hong Kong
[5] Finnish Meteorological Institute, Helsinki, Finland
[6] Department of Physics, University of Helsinki, Helsinki, Finland
[7] Department of Civil Engineering, the Chu Hai College of Higher Education, Castle Peak Bay, Hong Kong

*Correspondence to*: J.P.Wang (jiaping16@126.com); S.C.Lee (shun-cheng.lee@polyu.edu.hk)

**Abstract.** Temporal variations of aerosol optical properties were investigated at a coastal station in Hong Kong based on the field observation from February 2012 to February 2015. At 550 nm, the average light scattering (150.6 Mm$^{-1}$) and absorption coefficient (8.3 Mm$^{-1}$) were lower than most of other rural sites in eastern China while the single scattering albedo (SSA=0.93 ± 0.05) was relatively higher compared with other rural sites in the Pearl River Delta (PRD) region. Correlation analysis showed that the darkest aerosols were smaller in particle size but showed strong scattering wavelength dependencies, indicating possible sources from fresh emissions close to the measurement site. Particles with $D_p$ of 200-800 nm were less in number, yet contributed the most to the light scattering coefficients among submicron particles. In summer, both $\Delta BC/\Delta CO$ and $SO_2/BC$ peaked, indicating the impact of nearby combustion sources on this site. Multi-year backward LPDM and PSC analysis revealed that these particles were mainly from the air masses moved southward over Shenzhen and urban Hong Kong and the polluted marine air containing ship exhausts. These fresh emission sources led to low SSA during summer. For winter and autumn months, contrarily, $\Delta BC/\Delta CO$ and $SO_2/BC$ were relatively low, showing that the site was more under influence of well-mixed air masses from long-range transport including South China, East China coastal regions, and aged aerosol transported over Pacific Ocean and Taiwan Island, causing stronger abilities of light extinction and larger variability of aerosol optical properties. Our results showed that ship emissions in the vicinity of Hong Kong could have visible impact on the light scattering and absorption abilities as well as SSA at Hok Tsui.





## 1. Introduction

Atmospheric aerosol strongly affects the earth's radiative balance by scattering and absorbing incoming solar radiation, which however, is still a large source of uncertainties in global climate forcing assessment (IPCC, 2013). The aerosol optical properties are responsible for the direct aerosol radiative forcing, depending on aerosol chemical composition and

microphysical properties. Relative to another major component of radiative forcing, greenhouse gases, the shorter atmospheric lifetime of aerosols leads to more localized effects and regional differences in aerosol optical properties. Due to the spatial and temporal differences of aerosol optical properties caused by the complex distribution of tropospheric aerosols, field monitoring of aerosol optical properties in different regions around the world is critical for exploring the variations of aerosol radiative forcing. Among the major aerosol radiative forcing drivers, mineral dust, sulfate, nitrate and organic carbon generally have

negative radiative forcing. Contrarily, absorbing aerosols, like black carbon (BC), can strongly absorb visible light enhancing the warming effect of the atmosphere (Jacobson, 2001; Babu and Moorthy, 2001; Ding et al., 2016).

Light absorption and scattering of different kinds of aerosols have distinct wavelength dependencies that are approximately proportional to $\lambda^{-AAE}$ or $\lambda^{-SAE}$, respectively, where $\lambda$ is the wavelength and AAE and SAE are the Ångström exponents of

absorption and scattering, respectively. Hence, the wavelength dependency of aerosol light scattering and absorption has been recognized as an efficient index to distinguish aerosol types (e.g., Russell et al., 2010; Moosmüller and Chakrabarty, 2011; Devi et al., 2016). For instance, BC can strongly absorb light at all visible wavelengths, while other light absorbing aerosols (some organic aerosol, soil, and dust) absorb more blue light than red light (Moosmüller et al., 2011; Bond et al., 2013; Ding et al., 2016). Therefore, the absorption Ångström exponent (AAE) is often related to the dominant absorbing aerosol type for

a mixture of aerosols (Cazorla et al., 2013). the AAE in externally mixed BC-dominated regions have been reported to be around 1 (Anderson et al., 2007; Hegg et al., 2002; Bond and Bergstrom, 2006; Bond et al., 2013), while it is greater than 1 for some organic aerosol from biomass smoke and mineral dust due to their diverse light absorbing abilities at different wavelength ranges (Kirchstetter et al., 2004;Russell et al., 2010; Devi et al, 2016). Moreover, studies have shown that AAE of BC has a large variability depending on the size of BC cores and coating thickness (e.g., Lack and Cappa, 2010). For non-

coated BC with small diameter (e.g. 10 nm), AAE is close to 1, but large BC cores can have AAE<1 (e.g., Gyawali et al., 2009; Lack and Cappa, 2010). For coated BC particles, laboratory measurements of Schnaiter et al. (2005) reported that thickly coated BC by α-pinene plus ozone SOA could decrease the AAE to 0.8. Coating of BC by purely scattering material may also result in AAE up to about 1.8 (Gyawali et al., 2009; Lack and Cappa, 2010). The scattering Ångström exponent (SAE) is often regarded as a qualitative indicator of the dominating particle size, that is, large values (SAE > 2) indicate a large contribution

of small particles and small values (SAE < 1) a large contribution of large particles. For instance, Delene and Ogren (2002) reported that the influence of large sea-salt particles led to the lower SAE. However, this interpretation is not quite unambiguous, as was shown, e.g. by Schuster et al. (2006) and Virkkula et al. (2011). The single-scattering albedo (SSA) is the ratio of scattering to extinction coefficient, i.e., the sum of scattering and absorption coefficients. It equals 1 for purely





scattering aerosol and clearly lower, approximately 0.3 for pure BC particles (e.g., Schnaiter et al., 2003; Mikhailov et al., 2006). SSA varies significantly for smoke of different origin and age and correlates with the presence of BC in the combustion products (e.g., Dubovik et al., 2002).

There are several ways to assess the sources of aerosols, for instance by comparing observed particle concentrations with other tracers. As a by-product of the incomplete oxidation, CO has a long lifetime (about 1-2 months) in the troposphere, which therefore can act as a tracer of anthropogenic emissions (Jennings et al., 1996). A strong positive correlation between BC and CO has been found in previous studies concerning source identifications (Pan et al., 2011; Jennings et al., 1996). The BC/CO ratio is considered as a good indicator to determine BC emission and to recognize source characteristics. Also, the emission
ratio of BC and CO varies significantly from different sources, making it an effective index for validating emission inventories (Girach et al., 2014). The $SO_2$/BC ratio can be also used for assessing the sources since both BC and $SO_2$ are emitted in fossil fuel combustion (Bond et al., 2013).

The Pearl River Delta (PRD) region in Southern China has undergone fast industrialization with increasing emissions of
15 particulate and gaseous pollutants (Wang et al., 2003). In particular, the growing crisis of high particulate matter (PM) levels in the Pearl River Delta (PRD) region is of great concern due to its adverse effects on regional and continental atmospheric environment (Wang et al., 2009; Ding et al., 2013; Lam et al., 2005; Liu and Chan, 2002; Verma et al., 2010). Hong Kong is a typical coastal city in the PRD region. Under the influence of the East Asian monsoon, this region is controlled by the southerly winds bringing marine inflow from the South China Sea in summer, while in winter it is downwind from the North
and East China Plains and dominated by the continental outflow (Ding et al., 2013; Lam et al., 2001; Zhou et al., 2013). Thus, it is an ideal place for exploring the characteristics of optical properties for continental and marine aerosol.

There have been studies concerning aerosol optical properties and light-absorbing aerosols in the PRD region. Man and Shih (2001) did field observations of light scattering and absorption coefficients from September 1997 to April 1999 in Hong Kong.
Cheng et al. (2006a) investigated the seasonal variation patterns of BC concentrations in Hong Kong as well as the potential sources of BC by continuous measurement from June 2004 to May 2005 using model AE-42 Aethalometer (Magee Scientific Inc., Berkeley, Calif.). Cheng et al. (2008) presented the one-month record of aerosol optical measurements with related chemical apportionment at Xinken in PRD region and reported a relatively low SSA at this polluted rural site. Mixing states of light-absorbing aerosols were also investigated using optical closure experiments during campaign (Cheng et al., 2006b;
Tan et al., 2016). However, long-term observations of several key aerosol optical properties including wavelength dependencies of light scattering and absorption, single scattering albedo and studies on the relationships between optical properties and particle size, as well as their quantitative linkage to multi-scale transport are limited in Hong Kong over the past decade.





In this study, we aim at demonstrating the temporal variations of aerosol optical properties at a coastal station in Hong Kong and investigating the relationships between aerosol optical properties and size distributions based on field observations. Source analyses are conducted by comparing observed BC-to-CO ratios as well as the $SO_2$-to-BC ratios. Transport pattern and origins of aerosols were quantitatively studied based on backward Lagrangian particle dispersion modeling (LPDM). Characteristics

of local aerosol optical properties dominated by different aerosol source regions were also compared and illustrated.

## 2. Methodology

### 2.1 Sampling site

The Hok Tsui (HT) monitoring station is situated on the southeast tip of Hong Kong Island facing the South China Sea

(22.22° N, 114.25° E, 60 m above the sea level) with an almost vertical drop to the sea. This station has a view of the sea for over 180° from the northeast to southwest and is 20 km away from urban area of Hong Kong on the northwest. Owing to the characteristics of the location mentioned above, it is an ideal background monitoring site for identifying both the long-range transport of polluted continental/marine air mass caused by anthropogenic emissions and relatively clean marine air mass in different seasons. For more details about the HT site, please refer to Wang et al. (2009) and papers cited in.

### 2.2 Light absorption measurement

Light absorption measurement was conducted using a model AE-31 Aethalometer (Magee Scientific Company Berkeley, California, USA) from 1 February 2012 to 30 September 2013 and 1 March 2014 to 28 February 2015. Sample air was obtained through a stainless steel inlet with a $PM_{2.5}$ cut-off, protected with a rain cap. Prior to entering the instrument, sample air was

20 heated to ensure a moderate relative humidity. The sample inlet was approximately 1.5 m above the roof of the measurement station building, which was about 4 m above the ground. The sample flow provided by the internal pump was set to 4.0 LPM. The AE-31 Aethalometer performs continuous measurements of BC concentrations at seven wavelengths (370 nm, 470 nm, 520 nm, 590 nm, 660 nm, 880 nm and 950 nm) with a time resolution of 5 min. In this work, without specific notes BC concentrations refer to the aethalometer data measured at $\lambda$ = 880 nm. In order to correct the systematic errors of filter-based

absorption technique, the light absorption coefficients at all wavelengths were corrected using the method presented by Collaud Coen et al. (2010) where the $C_{ref}$ factor was set to be 4.26 according to the value from CAB station reported in the same paper. Absorption coefficients were presented under Standard Temperature and Pressure (STP) (273.15 K, 1013 hPa). Measured BC concentrations were corrected following the algorithm presented by Virkkula et al. (2007).

### 2.3 Light scattering measurement

Light scattering coefficients ($\sigma_{sp}$) at wavelength of 450 nm, 550 nm and 700 nm were measured using an integrating nephelometer (Model 3563, TSI Inc, St. Paul, MN, USA). The averaging time was set to 5 min. Calibration was conducted



once a month using $CO_2$ and filtered air as described in the user manual. An internal heater was used to maintain a moderate relative humidity during measurement. Raw $\sigma_{sp}$ data were corrected for truncation errors following the method from (Anderson and Ogren, 1998) where the scattering coefficients were determined by calculating the Ångström exponents from uncorrected scattering coefficients and the correction factors with no-cut inlet. Scattering coefficients were then corrected to STP using

pressure and temperature readings from the nephelometer.

### 2.4    Particle size measurement

An Ultrafine Particle Monitor (UFP, Model 3031, TSI Inc.) was used to measure the number size distribution of particles in the size range of 20 to 800 nm with six size bins of mobility diameter: 20~30 nm, 30~50 nm, 50~70 nm, 70~100 nm,

100~200 nm and 200~800 nm. The operating principle of UFP Monitor is based on diffusion charging of particles, followed by size segregation within a Differential Mobility Analyzer (DMA) and detection of the aerosol via a sensitive electrometer. The UFP monitor was equipped with a Model 3031200 environmental sampling system. The sample inlet was placed 2.0 m above the ground. Ambient air was continuously drawn through a size selective $PM_{10}$ inlet at a standard flow rate of 16.7 L/min. The sample then passed through a $PM_1$ cyclone to remove larger particles. The main sample stream was

subsampled into the UFP at a flow rate of 5 L/min. A Nafion dryer was installed upstream of the UFP to ensure proper conditioning of the aerosol and to minimize effects due to water vapor. The remaining 11.7 L/min of make-up air, drawn through a vacuum pump and exhausted, was routed through the Nafion dryer as purge air. The averaging time was set to 15 min.

The total mass concentrations of particles with mobility diameter less than 800 nm were calculated using the following equation:

$$m_{0.8} = \sum_{i=1}^{n} N_i \rho_i \frac{\pi}{6} D_{p,i}^3 , \qquad (1)$$

where $N_i$ was the number concentration in each size bin, $\rho_i$ was the density of particles assumed to be 1.7 g $cm^{-3}$, $D_{p,i}$ was the geometric mean of the upper and lower limit diameter in each size bin.

The size distributions were used for calculating scattering coefficients from:

$$\sigma_{sp}(\lambda) = \int Q_{sp}(\lambda, D_p, m) \frac{\pi D_p^2}{4} n(D_p) dD_p , \qquad (2)$$

where the scattering efficiencies ($Q_{sp}$) were calculated by using the BHMIE code (Bohren and Huffman, 1983). We assumed that the $D_p$ of each particle is equal to the geometric mean of the upper and lower limit diameter in its size bin for modeling, and the aerosol is ammonium sulfate with the refractive index $m = m_r = 1.52$ (Chamaillard et al., 2006). The refractive index

used in the modeling could in principle be varied and iterated until the measured and modeled scattering coefficients match as



was done, e.g., by Virkkula et al. (2011). However, due to the different size ranges and low number of size bins of the size distributions, this kind of iteration is not reasonable for the data in this work.

## 2.5 Supporting measurements

CO data was used to help analyzing aerosol sources since it typically originating from incomplete combustion like BC. Hourly mixing ratios of carbon monoxide was measured with a nondispersive infrared absorption instrument (Teledyne API Model 300) at Hok Tsui station.

In addition to the measurements at the HT station, the following supporting data measured at two near-by sites were used in the analyses. $SO_2$ is the precursor of sulphate, the most important light-scattering constituent and it is also one of the major pollutants of ship emission. $PM_{2.5}$ concentrations can be used for a semi-quantitative quality check of the aerosol mass concentrations calculated from the size distributions. Hourly $SO_2$ and $PM_{2.5}$ concentrations at Eastern station (about 7 km away from HT station, the location is shown in Fig. 1b) were downloaded from the open-access dataset from the website of Hong Kong Environmental Protection Department (HKEPD).

The hourly averaged meteorological parameters including air temperature, relative humidity (RH), wind direction, wind speed and precipitation were obtained from dataset in the HKEPD in which meteorological data from the nearest meteorological station (Waglan Island, WGL) was used for analyzing in this paper. The location of WGL station is shown in Fig. 1b.

## 2.6 Backward Lagrangian particle dispersion modeling (LPDM)

Transport and dispersion simulations were conducted using a Lagrangian particle dispersion modeling (LPDM) following the method developed by Ding et al. (2013). LPDM was conducted by using the Hybrid Single-Particle Lagrangian Integrated Trajectory (HYSPLIT) model, developed in the Air Resource Laboratory (ARL) of the USA National Oceanic and Atmospheric Administration (Draxler, 1998; Stein et al., 2015). In each simulation, particles were released at a height of 100 m above the ground level at the site and backward in time for a 7-day period. LPDM calculations were driven with GDAS (Global Data Assimilation System) data (http://ready.arl.noaa.gov/HYSPLIT.php). Particle positions were calculated in each hour and gridded concentrations were in a spatial resolution of 0.01° in latitude by 0.01° in longitude.

Knowing the transport characteristics of air masses, the next step was to explore the source profile of light absorbing particles affecting the regional aerosol optical properties in Hong Kong. Since BC is the most significant light-absorbing constituent of aerosols, the potential source contribution (PSC) of BC to observed air masses was calculated using MIX Asian emission inventory (Li et al., 2015) together with LPDM results. The MIX emission inventory has a horizontal grid resolution of 0.25°×0.25° in longitude and latitude and it considered the anthropogenic emissions from transportation, residential, industry





and power generation in continental area. In each grid cell, emission rate was multiplied with the footprint retroplume and the sum of this potential source contribution of all grid cells can provide the total BC concentration resulting from emissions during a certain period (Ding et al., 2013). The maps of averaged source contribution profile of BC in different seasons were calculated covering 70°-140° in longitude and 0°-50° in latitude. This method to calculate the PSC of target pollutants has

been adopted in a previous study by Ding et al. (2013). The major advantage of this method is that it captures the potential contribution of target pollutants to the receptor due to the transport of air mass containing the information of anthropogenic emissions.

In this study, the MIX emission inventory provided relatively high spatial resolution of BC emission rates considering its major

anthropogenic sources in China and nearby Asian countries. However, marine emission is not included in the MIX database. To investigate the possible influence of marine sources, like ship emissions, on the observed aerosol concentrations at this coastal site, we used the observed aerosol concentrations together with the LPDM footprint. We used the following concentration-weighted equation to calculate the potential source contribution from each grid cell:

$$A_{x(i,j)} = \frac{\sum_{t=1}^{n} (x_t \cdot R_{t(i,j)})}{\sum_{t=1}^{n} R_{t(i,j)}},$$ (3)

where x is the selected optical property or other parameters, and we chose $\sigma_{ap}$, $\sigma_{sp}$ and $m_{0.8}$ in this study. R represents the retroplume with 3-day backward time. $t$ is the time step and $n$ is the total number of the time steps. The interpretation of *Eq.(3)* is that it shows the average value of the property x observed at the receiving site when air masses have come from over grid cell i,j. The method is analogous to that presented by Stohl et al. (1996) and the concentration-weighted trajectory (CWT) methods reviewed by Cheng et al. (2015). The major difference is that in the present approach the footprints were used instead

of single trajectories for each time step.

## 3.  Results and discussions

### 3.1  Aerosol optical properties and their relationships with particle size

#### 3.1.1  Overall results of aerosol optical properties and related parameters

Table 1 shows a basic statistical summary of all measured parameters. The light absorption coefficients at λ=550 nm were interpolated between the $\sigma_{ap}$ at 520 nm and 590 nm. The mean absorption and scattering coefficients at λ=550 nm during the whole measurement period were 8.3 Mm$^{-1}$ and 146.9 Mm$^{-1}$, respectively. Table 2 summarizes the light scattering and absorption coefficients and single scattering albedos observed in this study and in selected other studies using comparable instruments (Man and Shih, 2001; Xu et al., 2002; Yan et al., 2008). On average, the $\sigma_{ap}$ was lower than that measured at



Lin'an regional background station in the rural area of the Yangtze River Delta Region (Xu et al., 2002). Compared to the value measured at same station, $\sigma_{ap}$ was lower than that observed in Hok Tsui from November 1997 to February 1999 (Man and Shih, 2001). As the most significant light-absorbing constituent of aerosols, a similar decrease of BC concentration was also found. Table 3 presents the mean BC mass concentrations reported in other comparable studies. The overall average of BC mass concentrations in this study was 1.4 µg/m³ (Table 1), which was lower than the values observed at same site in 2004-2005 (with a mean of 2.4 µg/m³ using AE-42 Aethalometer) (Cheng et al., 2006). A decreasing trend of BC concentration was found at Panyu station in the PRD region with a decreasing rate of approximately 1 µg/m³ per year from 2004 to 2007 (Wu et al., 2009). Compared to the other rural sites in the South China, BC levels in Hok Tsui station were lower than the concentrations measured at a rural site in the center of PRD region, yet higher than those on Yongxing Island, an oceanic rural site in the middle of the South China Sea (Yu et al., 2013). BC concentrations were also higher than those measured in European coastal stations (Saha and Despiau, 2009; Andriejauskienė, 2008). The $\sigma_{sp}$ was comparable to that obtained at Shangdianzi station in the suburb of Beijing, but much higher than the value at Hok Tsui station measured a decade ago (Yan et al., 2008; Man and Shih, 2001). The overall average $SSA_{550nm}$ was 0.93, which was comparable to that in a rural station, Lin'an, China (Xu et al., 2002) but higher than those measured in a suburban station in Northern China (Mean $SSA_{525nm}$ =0.88) (Yan et al., 2008). Compared with other rural/background sites, $SSA_{550nm}$ at Hok Tsui was higher than that measured at Xinken, PRD, China during autumn (0.83 ± 0.05) but slight lower than that observed at a coastal station in Norway in summer (0.91 ± 0.05, Mogo et al., 2012). CO mixing ratios in Hok Tsui station were comparable to those measured at same site in 1994-1996 (Lam et al., 2001).

### 3.1.2 Temporal variations and overall characteristics

The seasonal cycles of target parameters were analyzed based on hourly-averaged data classified as four seasons: winter (December-February), spring (March–May), summer (June–August), and autumn (September–November). Seasonal averaged values of selected parameters are listed in Table 4. The highest $\sigma_{ap}$ and $\sigma_{sp}$ values were observed in winter (10.9 Mm⁻¹ and 193.5 Mm⁻¹, respectively), which were more than twice that of summer. Similar pattern was observed in a previous study in Hong Kong in 1997-1999 (Man and Shih, 2001).

Fig. 2 presents the monthly variation of measured optical properties and meteorological parameters. A clear seasonal cycle of aerosol optical properties is shown with $\sigma_{ap}$ and $\sigma_{sp}$ peaked in January and reached to the lowest level in July. The aerosol was the darkest in summer especially in August, with a seasonal mean SSA of 0.87, while it was lighter in winter (average SSA = 0.94). Averaged seasonal values of 1-SSA in 36 wind sectors are presented in Fig. 3a. These figures show the disparity of single scattering albedo from different wind directions. Overall, air plume coming from the southwest to the north (225-360°) had higher 1-SSA, i.e. lower SSA, than that from the east (45-135°). Ding et al. (2013) reported that the contribution of anthropogenic emissions from Guangdong and Hong Kong was the highest in August, which means more freshly emitted urban aerosols were brought to the monitoring station with lower SSA in this month (Cheng et al., 2008). Main synoptic process contributing to this kind sub-regional transport is tropical cyclones. Ding et al. (2004) explained the mechanism on



how these tropical cyclones influence the development of sea-land breeze and further on sub-regional and urban air mass accumulation in the South China. Zhang et al. (2013) found an important influence of tropical cyclones in ozone and haze pollution in this region in summer based on an analysis of 13-year data.

Another possible reason for the relatively low SSA in August is that the air mass came mainly from the southwest of the site (Fig. 1), a main waterway for ocean-going vessels in Hong Kong (Yau et al., 2012). These vessels emitted considerable amount of light absorbing carbon from diesel engines during combustion. Similar pattern was also observed in the seasonal diagrams of BC, $SO_2$, $PM_{2.5}$ and CO which are typical components of ship exhaust (Fig. 2, Hong Kong Air Pollutant Emission Inventory for 2013 from Hong Kong Environmental Protection Department: http://www.epd.gov.hk/epd/english/environmentinhk/air/data/emission_inve.html).

Fig. 3b demonstrates the averaged 7-day retroplume of the times when SSA was lower than 0.9. Compared with the overall averaged 7-day retroplume during the whole measurement period (Fig. 3c), darker aerosols were mostly from two main types of regions in the vicinity: one was the nearby continental area, where fresh polluted air masses from urban Hong Kong and neighboring PRD cities, another branch was from the ocean side. Fresh emission of passing ships or fast transport from the South Asia could lead to higher proportion of BC in the air plumes and thus caused lower SSA.

Fig. 4 shows the diurnal cycles of $\sigma_{ap}$, $\sigma_{sp}$, BC, CO and $m_{0.8}$ for four seasons. There was an increase of $\sigma_{ap}$ after sunrise with peak occurred before noontime. It might be associated with a combined effect of increased human activities and turbulence mixing in the boundary layer in the morning. This pattern was more significant in summer although the pollution level was relatively low. This phenomenon supports the explanation of turbulent mixing from middle or upper planetary boundary layer (PBL) because of a stronger vertical mixing in summer. The $m_{0.8}$ also showed a daytime maximum concentration but with the peak in the afternoon (Fig. 4c). For $\sigma_{sp}$, morning peaks were not as significant as $\sigma_{ap}$. The decrease of $\sigma_{ap}$ in the early afternoon might cause by a further development of PBL or mixing layer, in which the air pollutants experienced a substantial dilution, resulting in lower concentrations of pollutants at the ground surface. Diurnal variations and fluctuations of CO mixing ratios show a similar pattern with $\sigma_{ap}$ but a relatively smaller variability.

### 3.1.3 Optical properties and their relationships with particle size

Wavelength dependencies of aerosol light scattering and absorption are closely related to aerosol size and dominating aerosol types. To find out the difference of light absorbing materials, Fig. 5a displays the relationship of SSA with AAE color-coded with BC mass fraction of submicron particles ($m_{0.8}$ was calculated from the particle number size distributions measured with the UFP monitor). It shows that aerosols with high SSA had lower BC fraction and that AAE varied greatly in the lower value region, indicating the dominance of scattering particles. Such kind of air masses was likely of longer transport time and the BC aerosols had mixed well with light scattering aerosols during transport. Contrarily, the low SSA values mostly occurred when AAE were closely distributed around 1.0 and in these cases BC took up a higher proportion (red dots in Fig. 5a), showing a fresh-emitted BC plumes.



Fig. 5b and 5c demonstrate the relationships between particle size and their scattering Ångström exponents as well as their darkness. It can be observed that SAE generally increased with decreasing SSA. Dark aerosols with low SSA were mostly small in size with low GMD but high BC fraction. These small dark aerosols had higher SAE (1.5 to 2.0). The wide range of SAE was possibly due to the mixed control by continental aerosols and large sea-salt aerosols.

Fig. 6 shows the scatter plot of $\sigma_{sp}$ calculated using *Eq.(2)* versus the measured $\sigma_{sp}$. The slope of $\sigma_{sp,\ submicron}\ /\ \sigma_{sp,\ obs}$ was 0.86, indicating that submicron particles were the major light scattering components in the air masses arriving at the Hok Tsui station. For most of time in the study period, the simulated $\sigma_{sp}$ was lower than the observed $\sigma_{sp}$. This is probably because that the particle size distribution data from UFP monitor used in the calculation only the scanned submicron particles with mobility diameter from 20 to 800 nm (see Fig. 6b and 6c), but the nephelometer, equipped with a TSP inlet, measured light scattering coefficients from all particles with a wider size range. The relatively limited number of particle size bins in the UFP monitor probably also leads to uncertainties for the calculation of $\sigma_{sp}$. Hence, this result can only provide rough image of the relationships between particle light scattering and their size distribution at the Hok Tsui station. It can be observed that particles with $D_p$ less than 200 nm contributed the largest fraction of the total number of submicron particles but very little to the total scattering whereas the small amount of larger particles ($D_p$: 200-800 nm) contributed the most to the total light scattering.

The scatter plot (Fig 6a) also shows that there were clusters of data where the modeled and measured $\sigma_{sp}$ fit close to the 1:1-line and clusters where the measured $\sigma_{sp}$ was clearly larger than the modeled on. The former data cluster is most probably associated with polluted continental air and the latter with stronger winds and sea salt particles.

### 3.2    Source identification

Fig. 7 shows the spatial distribution of BC/CO emission ratios in East China and the nearby regions calculated using the MIX emission inventory. It can be seen that BC/CO emission ratio was higher in Shanxi Province (higher than 25 ng/m³/ppbv), Taiwan Island (approximately 20 ng/m³/ppbv) and the regions along the coastline of East China. As reported by previous studies, BC/CO emission ratio from industrial coal burning ranges from 1.9 to 20 ng/m³/ppbv and it was 5.6~13.3 ng/m³/ppbv from open biomass burning (Wang et al., 2011; Zhang et al., 2009). For diesel vehicles, BC/CO emission ratio was 14~39 ng/m³/ppbv and it was 15.6 ng/m³/ppbv for ship emission calculated from a previous study in South Asia (Dickerson et al., 2002). A strong correlation between BC and CO and a high slope of $27 \times 10^{-3}$ g BC/g CO were found from a previous study using C-130 aircraft flew over the Arabian Sea and Northern Indian Ocean (Dickerson et al., 2002; Mayol-Bracero et al., 2002).

In this study, $\Delta BC/\Delta CO$ and $SO_2/BC$ ratios were investigated to study the source characteristics and the freshness of the fuel combustion sources. $\Delta BC/\Delta CO$ (net growth of BC and CO: total concentration minus regional baseline, Spackman et al., 2008) and $SO_2/BC$ were calculated with 1-hour time resolution. The baseline of BC and CO were determined as 1.25th percentiles of data in each month (Pan et al., 2011). Monthly variation of $\Delta BC/\Delta CO$ is displayed in Fig. 8 together with $SO_2/BC$ to demonstrate the fuel burning emission profile since $SO_2$ is a co-emitted species of fossil fuel combustion (Bond et al., 2013).



Reference emission ratios of BC/CO and $SO_2$/BC from previous studies (Bond et al., 2013; Li et al., 2015) are also plotted in Fig. 8.

In Hong Kong, major $SO_2$ emission was from navigation and public electricity generation, contributing 50% and 47% to total $SO_2$ emission (Emission Inventory 2013, HKEPD,

http://www.epd.gov.hk/epd/english/environmentinhk/air/data/emission_inve.html). However, these two sources only took up 19% and 6% of CO emission and the largest contributor of CO reported in the emission inventory was road transport (59%). As shown in Fig. 8, $\Delta BC/\Delta CO$ and $SO_2$/BC ratios presented similar monthly variation patterns. The monthly mean $\Delta BC/\Delta CO$ ranged from 1.5 to 20 $\mu g\cdot m^{-3}$/ppbv during whole study period. The highest values occurred in summer months for both $\Delta BC/\Delta CO$ and $SO_2$/BC and the ratios were relatively lower in winter. Since $SO_2$ has short lifetime, which can easily deposit

and transform into secondary aerosols, the synchronous elevation of $\Delta BC/\Delta CO$ and $SO_2$/BC in summer indicates that freshly emitted anthropogenic pollutants might be more easily influenced by the air masses in this coastal area. The decrease of $\Delta BC/\Delta CO$ and $SO_2$/BC in winter provided the evidence that this area was under the influence of contaminated air masses from a longer distance. Fig. 9 displays the scatter plot of BC vs. CO in four wind sectors, giving an image of the freshness of polluted air masses and the intensities of combustion emissions from different directions. For wind directions from 180° to 360°, the

data points show a good positive correlation, suggesting that most of the BC and CO emission sources in these areas were closer to the measurement site. Data points in the 0-180° wind direction sector were much more scattered with a lower slope of BC-CO and weaker correlation coefficients, indicating the higher complexity of source regions and longer transport age of air pollutants coming from the northeast to southeast. Through the transport of air plume, diffusion and deposition of air pollutants would decrease their concentrations arriving to the receptor and therefore lower the slope of $\Delta BC/\Delta CO$ and their

correlation coefficients.

To investigate the transport pattern of air masses arrived at the site during the study period, Fig. 10 shows averaged 7-day retroplume for four seasons. As that presented in Ding et al. (2013), it shows a distinguished different transport patterns under the influence of Asian monsoon. During summer, the majority of air masses came from the south and nearby PRD cities. Due to the dominance of relatively clean marine air, emission from passing ships or local activities in adjacent regions could make

visible effects on the temporal variations of air pollutants. The source distribution was more complicated during winter. Driven by the winter monsoon, cold and dry air masses transported along the coastline of the East China and from central China took up a higher proportion in winter months (Ding et al., 2013). There were also air masses passing through Taiwan Island and the East China Sea during cold season (Fig. 10a).

Since BC is the most significant light-absorbing constituent of aerosols, to evaluate the potential source contribution of light absorbing particles on regional optical properties, averaged PSC maps of BC for different transport time and seasons were calculated using the method described in Sect. 2.6, and illustrated in Fig. 11. However, here we only calculated the PSC from emission over land because the available emission inventory from MIX database is mainly focus on land area. As shown in Fig. 11a and 10b, BC concentrations were influenced by the transport from nearby cities within a short time, especially



Shenzhen and urban Hong Kong. Long-range transport of BC from the South and East China also played important contribution. It was also showed that BC coming from continental area through longer distance took up a higher proportion of the pollutant level in winter (Fig. 11d) than that in summer month (Fig. 11c). During summer, local emission was the biggest BC contributor.

Fig. 12a-11c illustrates the average levels of $\sigma_{ap}$, $\sigma_{sp}$ and $m_{0.8}$ and the corresponding frequency of occurrence for air masses passing through different regions calculated using *Eq.(3)*. Together with the shipping routes density map (Fig. 12d), it can be observed that the high levels of $\sigma_{ap}$ and $\sigma_{sp}$ were closely associated with the congested shipping lanes in the maritime space nearby Hong Kong. The high $\sigma_{ap}$ and $\sigma_{sp}$ were especially visible in the northeast due to the prevailing northeasterly wind from autumn to spring, transporting ship exhausts mainly through the Taiwan Strait. The belt-like zone with higher $\sigma_{ap}$ and $\sigma_{sp}$ was

likely to be the reveal of ship emission. As shown in Fig. 12d, there were dense shipping routes between Hong Kong and Singapore transporting through the Xisha Islands in the South China Sea where the routes were similar to the high $\sigma_{ap}$ area in the south. During summer, Hong Kong was influenced by the southerly and southwesterly wind, bringing clean marine air to this region for the most of time and leading to the lower pollutant levels (Wang et al., 2009; Ding et al., 2013). Here Fig. 12 indicates that Hong Kong could be affected by the passing vessels in the South China Sea due to controlling wind direction

driven by the summer monsoon.

### 3.3 Analyses of selected episodes

Fig. 13 demonstrates the aerosol optical properties and BC-CO correlations associated with air masses from different source regions during selected episodes. The major source regions were Guangdong and Hong Kong (GH), ship emission (SP), North

China (NC), and aged continental area (AGC). The selection of the episodes was done by combining the footprints using LPDM and the variation trend of target parameters. The air pollution plumes coming from Guangdong and urban Hong Kong had the highest BC and CO concentrations (Fig. 13a), indicating higher level of emission intensity and stronger light extinction ability of aerosols from these regions. The slope of BC vs. CO was the highest from ship emission (0.012 µg/m$^3$/ppbv) with high correlation ($r^2$=0.84), showing that ship emission source was close to the measurement station and its exhausts could

largely affect the pollution level.

Fig. 13b displays that Ångström exponents of scattering from Guangdong and Hong Kong were relatively high as well as ship emission compared with that from aged continental area and North China, proving the dominance of smaller particles of emissions from PRD cities and passing ships. BC-containing particles transported from the North and East China went through

longer coating and deposition processes, which enlarged their size but decreased their concentrations arriving to the measurement site. This can further explain the lower SSA in summer months.

Overall, the analyses suggest that aerosols from different source regions could make great discrepancies on regional aerosol optical properties. Thus, more ground observations of aerosol optical properties are needed to fully understand the





characteristics of different types of atmospheric aerosols and provide reference datasets for further investigating aerosol radiative forcing and climatic effects.

## 4.  Conclusions

Based on aerosol optical properties, relevant species and aerosol size measured at Hok Tsui station in Hong Kong, we studied the temporal variations and investigated the potential sources by using correlation analysis and Lagrangian dispersion modeling. Overall, the absorption coefficients at the site in the South China coastal region were lower than most of other rural sites in eastern China. Scattering coefficients observed in this study were almost twice as the values monitored at the same station in 1998, yet BC concentrations decreased over fifty percent compared with the measurements in 2004. The darkest aerosols were
smaller in particle size but showed strong scattering wavelength dependencies, indicating possible sources from fresh emissions close to the measurement site. Particles with $D_p$ of 200-800 nm were less in number, yet contributed the most to the light scattering coefficients among submicron particles.

A remarkable correlation was found for BC and CO concentrations during episodes. $\Delta BC/\Delta CO$ range from
1.5 to 20 $\mu g \cdot m^{-3} \cdot ppb^{-1}$ during whole period. Both $\Delta BC/\Delta CO$ and $SO_2/BC$ peaked in summer months and were relatively low in spring and autumn. In summer, the site was affected by nearby combustion sources, while in spring and autumn the observed air masses were more under influence of well-mixed air masses from long-range transport. Multi-year backward LPDM and PSC analysis together with case studies provided detailed information of the transport of air masses and their impacts on aerosol optical properties. For summer months, air masses moved southward over Shenzhen and urban Hong Kong brought
air pollutants mainly from residential and transportation to the measurement site showing strong light absorbing ability, and ship exhausts were introduced into the southerly marine air with higher speed, showing strong positive correlation between BC and CO. These fresh emission sources led to low SSA during summer. For winter and autumn months, the air plume arriving at Hok Tsui station was a mixture of multi-source aerosol including air masses from South China, East China coastal regions, and aged aerosol transported over Pacific Ocean and Taiwan Island, causing stronger abilities of light extinction and
larger variability of aerosol optical properties as well as pollutant concentrations.

**Acknowledgements:**
The data measurement of this study was supported HK project (H-ZDA8), Research Grants Council of Hong Kong Government (PolyU 152083/14E, PolyU 152090/15E) and Hong Kong RGC Collaborative Research Fund (C5022-14G). The
data analysis was supported by NSFC (D0512/91544231), the National Special Research Fund for Non-Profit Sector (Environmental Protection) (No. 201509004). The authors would like to thank Mr. Steven Poon for the support of trace gases measurement at the Hok Tsui Site and thank Ding Ke for useful discussions.





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





**Figure Captions**

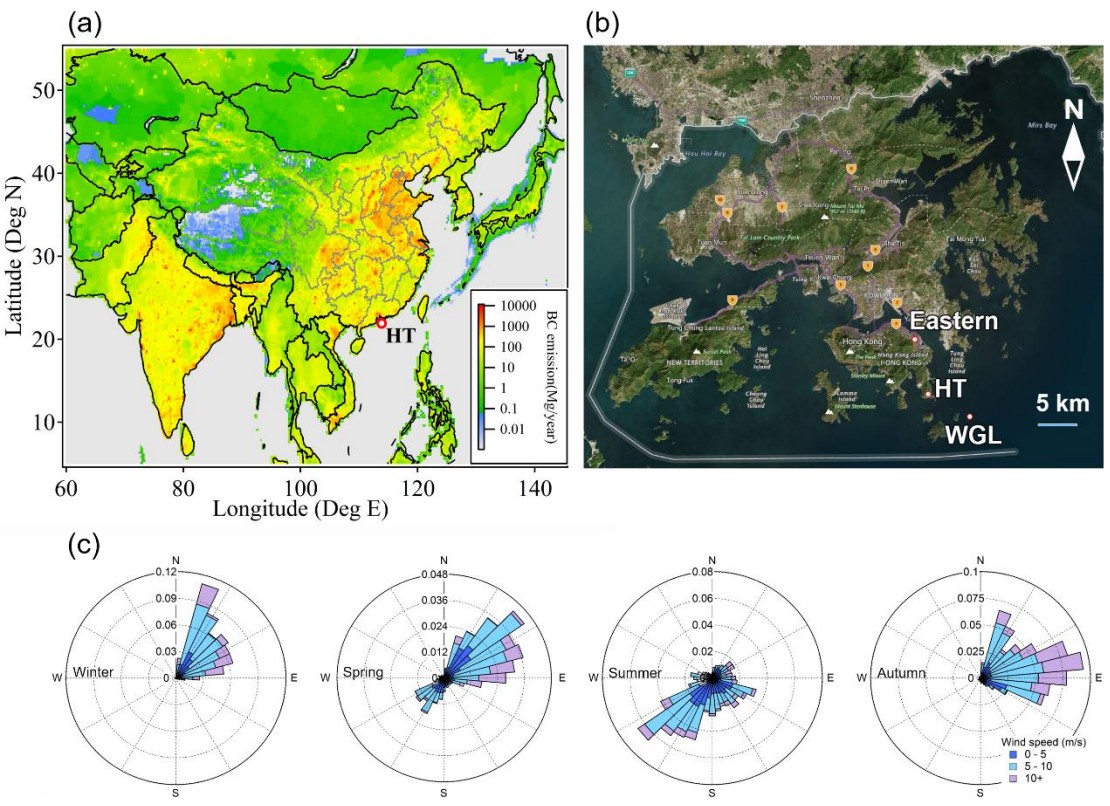

**Fig. 1. (a) Map showing the location of Hok Tsui (HT) monitoring station with emission inventory in Asia, (b) locations of monitoring stations mentioned in this paper and (c) wind rose plot at WGL in Hong Kong**



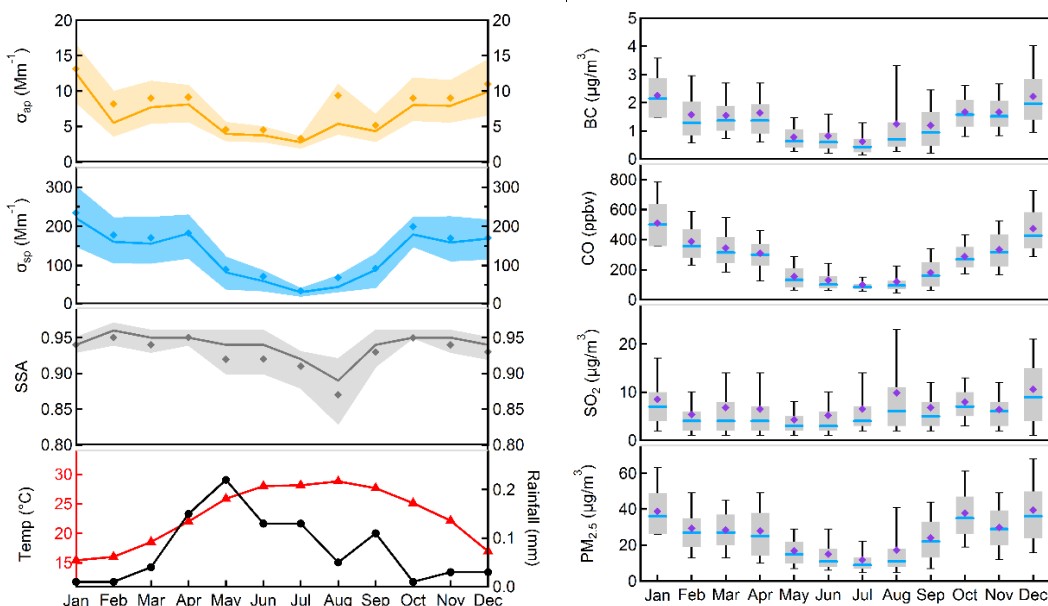

**Fig. 2.** Left: seasonal cycle of scattering coefficient, $\sigma_{sp}$, absorption coefficient, $\sigma_{ap}$, single-scattering albedo, SSA, temperature and precipitation where bold solid lines represent median values, diamonds show the monthly averages and thin solid lines are percentiles of 75 % and 25 %. Right: seasonal cycles of BC, CO, SO$_2$ and PM$_{2.5}$ concentrations, where blue solid lines represent median values, diamonds show the monthly averages, the boxes are 25[th] and 75[th] percentiles and the thin bars represent 10[th] and 90[th] percentiles





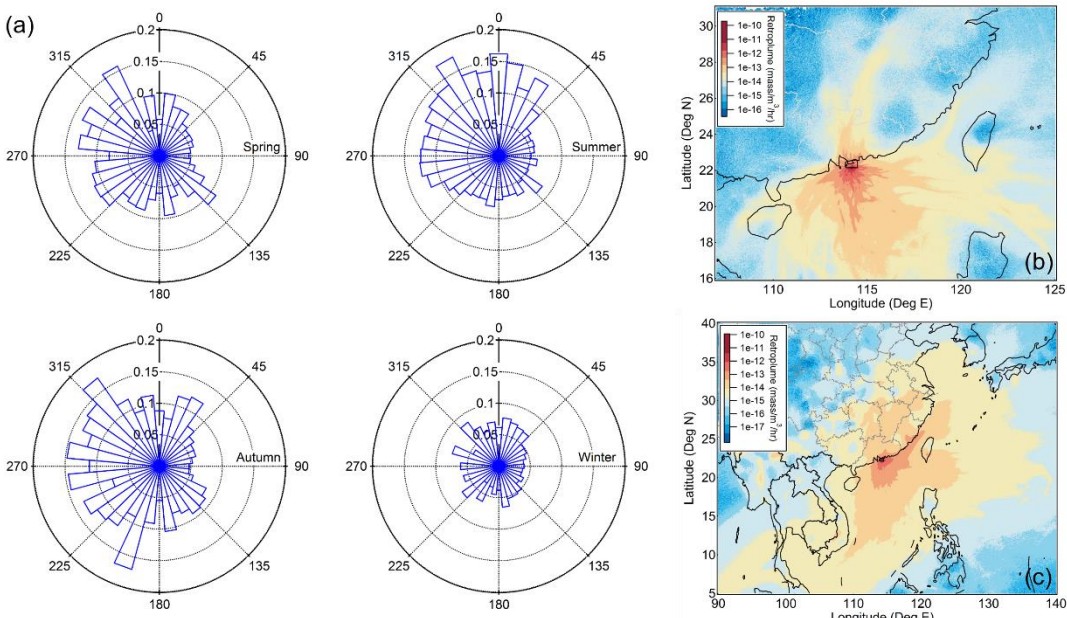

Fig. 3. (a) Seasonal mean value of (1-SSA) in 36 wind sectors during the whole period, (b) map of averaged 7-day retroplume when SSA is below 0.9 compared with (c) averaged 7-day retroplume during the whole period





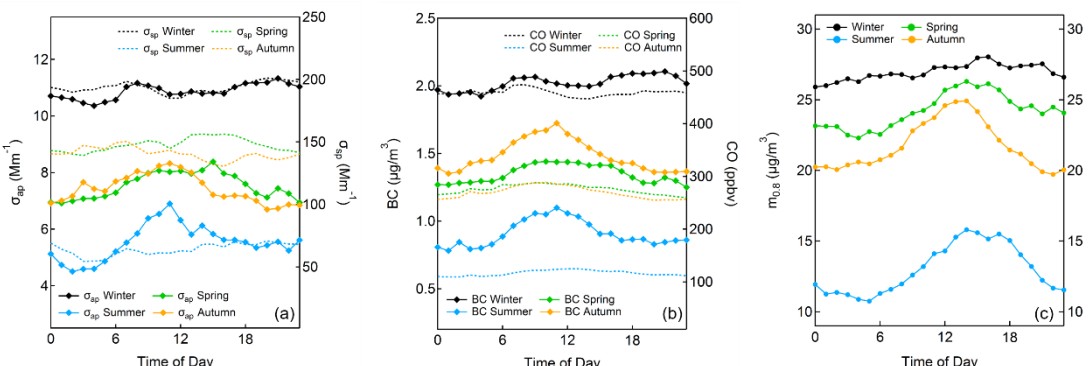

**Fig. 4. Averaged diurnal variations of (a) $\sigma_{sp}$ and $\sigma_{ap}$, (b) BC and CO, (c) $m_{0.8}$ in four seasons**





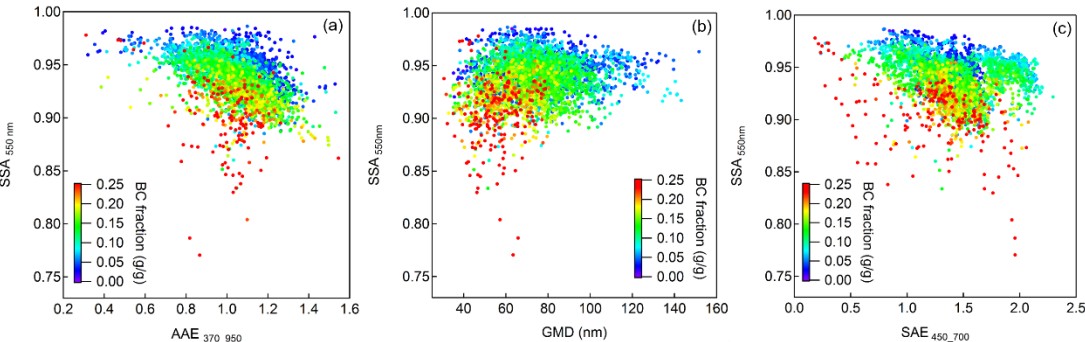

**Fig. 5. (a) Scatter plot of SSA$_{550nm}$ and AAE, (b) SSA$_{550nm}$ and GMD, and (c) SSA$_{550nm}$ and SAE$_{450\_700nm}$, color coded with BC mass**
**fraction of m$_{0.8}$**





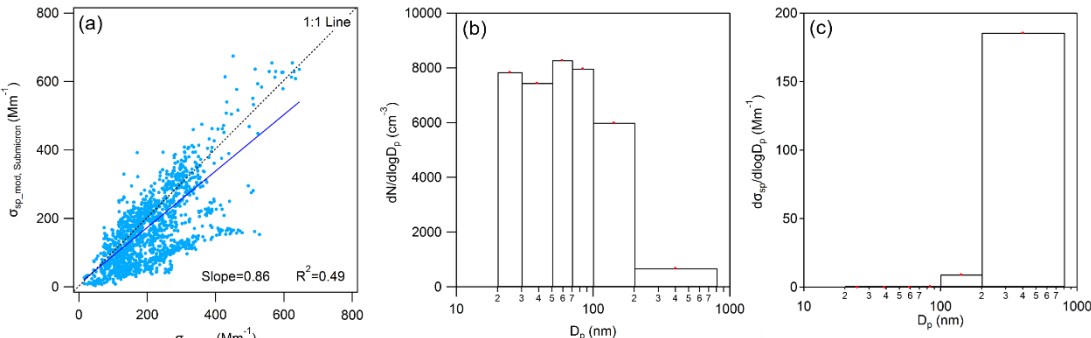

**Fig. 6.** (a) Scatter plot of simulated $\sigma_{sp}$ of submicron particles and observed $\sigma_{sp}$ at $\lambda=550$ nm, (b) average number size distribution and (c) scattering size distribution during the whole period.



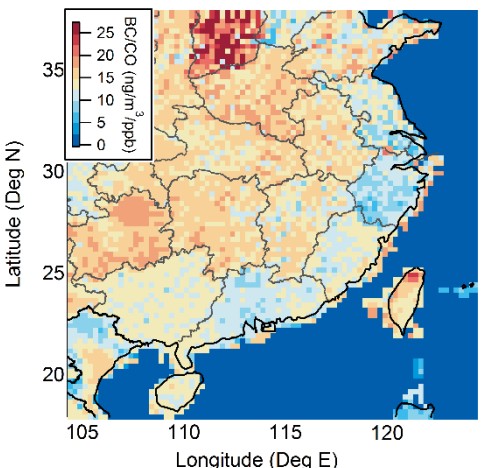

**Fig. 7 Map showing spatial distribution of BC/CO emission ratio with grid resolution of (0.25°×0.25°) from MIX Asian emission inventory (Li et al., 2015)**





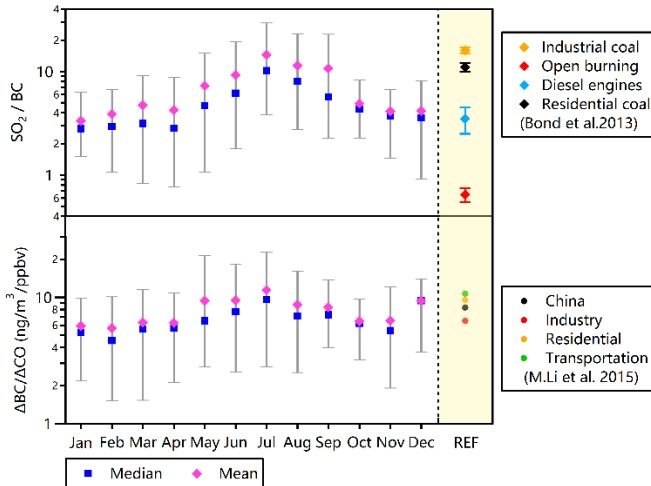

**Fig. 8. Seasonal cycles of ΔBC/ΔCO and SO₂/BC ratios from observations (reference values of emission ratios from different source types were shown in the column with light yellow background)**





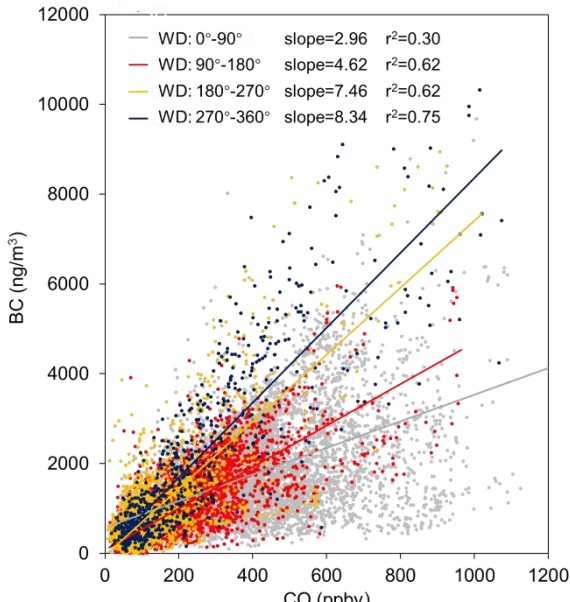

**Fig. 9. Scatter plot of hourly BC and CO in four wind sectors**





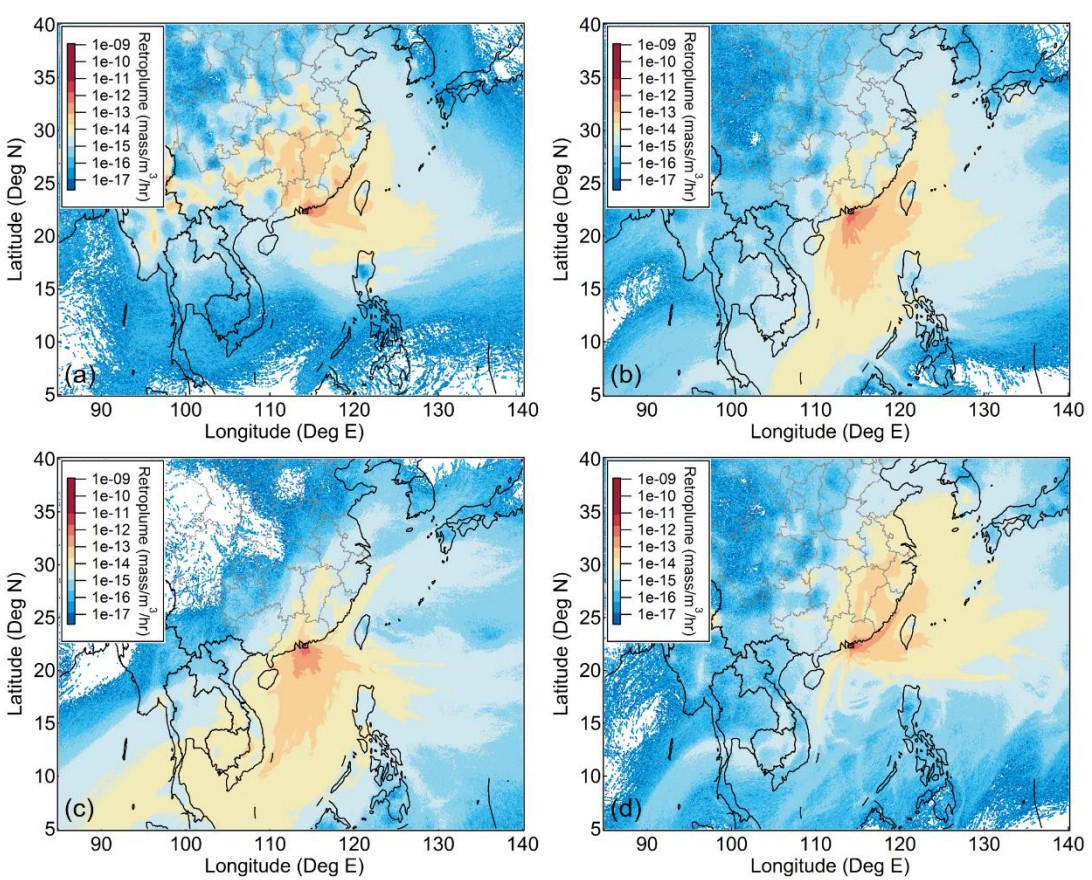

**Fig. 10. Map of averaged 7-day retroplume in (a) Winter (b) Spring (c) Summer (d) Autumn**





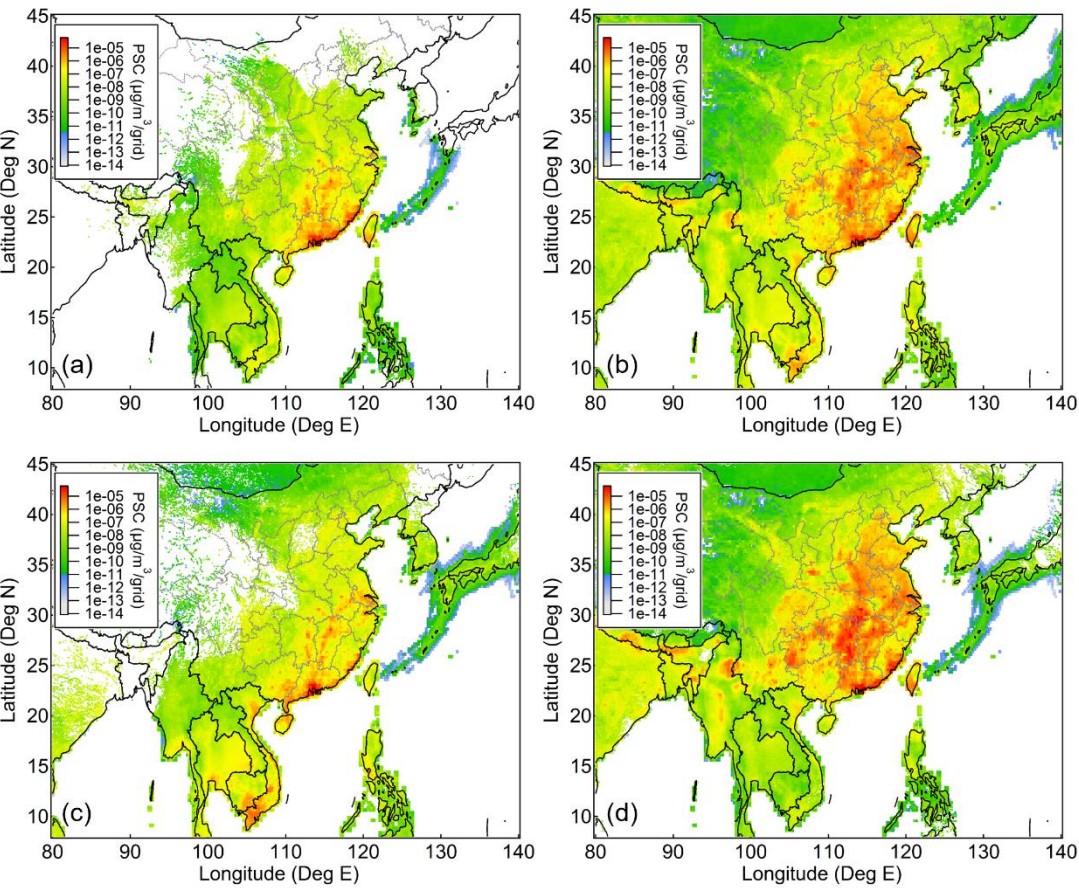

**Fig. 11 (a) (b): Map of averaged potential source contribution of BC with a backward transport age of (a) 2 days, (b) 7 days for the whole measurement period; (c) (d): averaged 7-day PSC in (c) Summer, (d) Winter**

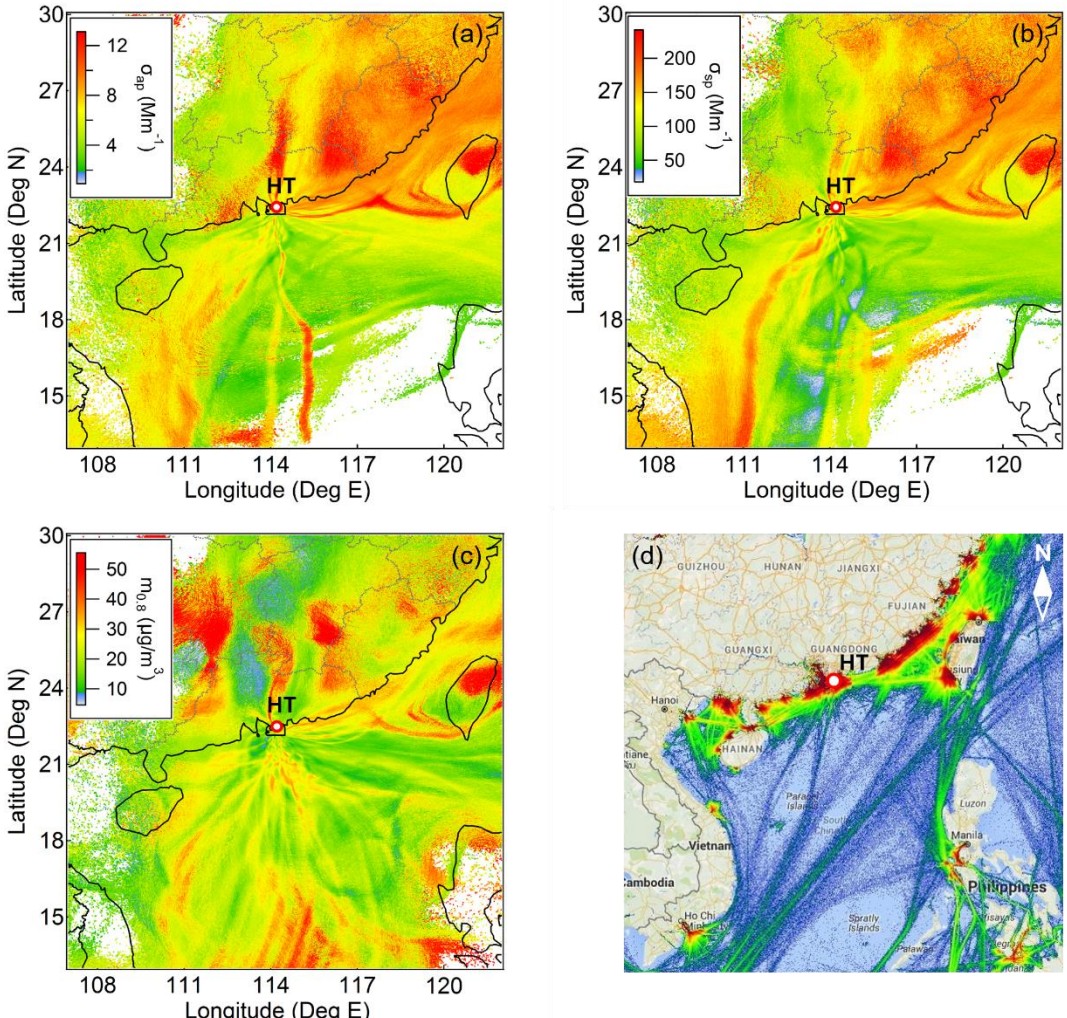

**Fig. 12.** Map of average property$_{\text{retroplume}}$ for (a) $\sigma_{ap}$, (b) $\sigma_{sp}$ and (c) $m_{0.8}$ (the non-colored areas were where the total retroplume was smaller than $10^{-12}$ mass/m³/hr (i.e., air plumes barely passed through these regions). Due to the different time period of valid data from UFP, the non-colored areas were slight different in (c)); (d) density map showing the ship routes near Hong Kong during 2013 and 2014.





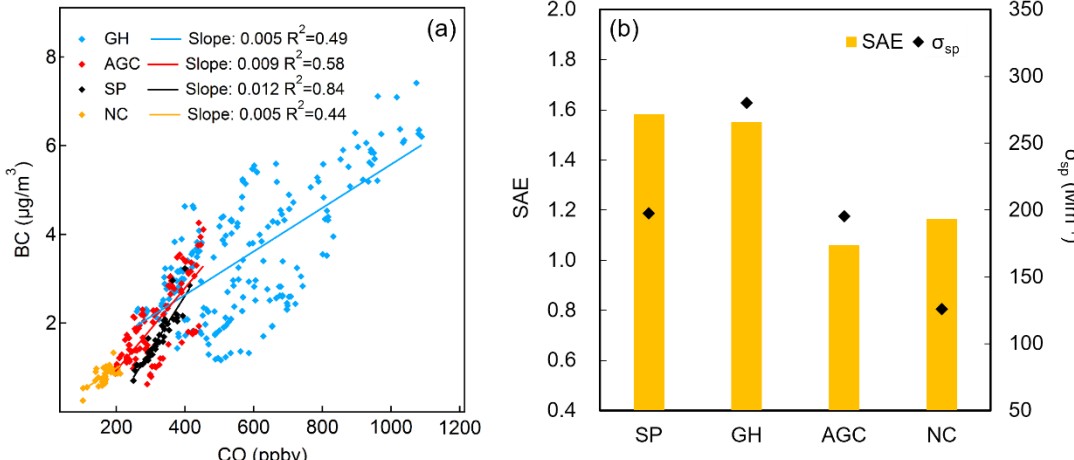

**Fig. 13.** **(a) Scatter plots of BC and CO, (b) $\sigma_{sp}$ and SAE from different source regions during episodes (GH: Guangdong and Hong Kong, SP: Ship, NC: North China, AGC: aged continental area)**





**Table 1. Statistical summary of data measured at Hok Tsui station. Scattering coefficients ($\sigma_{sp}$) and absorption coefficients ($\sigma_{ap}$) at λ=550 nm corrected to STP (1013 mbar, 273.15 K), Ångström exponents of scattering and absorption (SAE, AAE), single-scattering albedo (SSA), total particle number concentration ($N_{total}$), geometric mean diameter (GMD) and mass concentration of particles smaller than 0.8 μm ($m_{0.8}$).**

|  | AVG ± STD | Percentile | | | | |
|---|---|---|---|---|---|---|
|  |  | 5 | 25 | 50 | 75 | 95 |
| $\sigma_{ap, 550nm}$ (Mm$^{-1}$) | 8.3 ± 6.1 | 2.2 | 4.0 | 6.6 | 11.0 | 19.3 |
| $\sigma_{sp, 550nm}$ (Mm$^{-1}$) | 150.6 ± 99.4 | 23.4 | 74.8 | 134.3 | 205.6 | 331.4 |
| SSA (550nm) | 0.93 ± 0.05 | 0.84 | 0.92 | 0.94 | 0.96 | 0.98 |
| BC (μg/m$^3$) | 1.4 ± 1.1 | 0.2 | 0.6 | 1.2 | 1.9 | 3.4 |
| CO (ppbv) | 271.6 ± 185.4 | 59.1 | 108.8 | 242.3 | 373.5 | 623.3 |
| AAE | 1.04 ± 0.25 | 0.51 | 0.93 | 1.07 | 1.19 | 1.39 |
| SAE | 1.35 ± 0.45 | 0.62 | 1.10 | 1.36 | 1.60 | 2.04 |
| $N_{total}$ (#/cm$^3$) | 7789 ± 4302 | 2984 | 5063 | 6829 | 9413 | 15713 |
| GMD (nm) | 67.2 ± 17.4 | 43.1 | 54.7 | 65.3 | 77.2 | 98.4 |
| $m_{0.8}$ (μg/m$^3$) | 22.1 ± 19.1 | 3.1 | 9.6 | 18.0 | 28.9 | 54.7 |





**Table 2. Summarization of aerosol light scattering coefficients, absorption coefficients and single scattering albedo observed in this study and reported in other studies.**

| Site | Period | $\sigma_{ap}$ | $\sigma_{sp}$ | SSA | Instrumentation | References |
|---|---|---|---|---|---|---|
| Cape D'Aguilar, Hong Kong (rural, coastal) | Feb. 2012-Feb. 2015 | 8.3 | 150.6 | 0.93 | AE31, Magee Scientific Nephelometer, TSI, Inc. | This work |
| Cape D'Aguilar, Hong Kong (rural, coastal) | Nov. 1997-Feb. 1998 | 25.72 | 64.77 | | PSAP, Radiance Research Nephelometer, Radiance Research | (Man and Shih, 2001) |
| | Mar. –Apr. 1998 | 15.79 | 38.65 | | | |
| | May. –Aug. 1998 | 6.03 | 8.71 | | | |
| | Sep. –Oct. 1998 | 18.98 | 70.91 | | | |
| | Nov. 1998-Feb. 1999 | 31.22 | 96.75 | | | |
| Xinken, PRD, China (non-urban, regionally polluted) | Oct.-Nov. 2004 | 70 | 333 | 0.83 | MAAP, Thermo, Inc. Nephelometer, TSI, Inc. | (Cheng et al., 2008) |
| Shangdianzi, China (rural) | Sep. 2003–Jan. 2005 | 17.54 | 174.6 | 0.88 | AE31, Magee Scientific Nephelometer, EcoTech | (Yan et al., 2008) |
| Lin'an, China (rural) | Nov. 1999 | 23 | 353 | 0.93 | PSAP, Radiance Research Nephelometer, Radiance Research | (Xu et al., 2002) |
| La Valette, France (urban, coastal) | Oct. 2005-Oct. 2006 | 22 | 60 | 0.77 | AE31, Magee Scientific M9004 Nephelometer, EcoTech | (Saha et al., 2008) |
| ALOMAR station, Norway (background, coastal) | Jun.-Aug. 2008 | 0.4 | 5.41 | 0.91 | PSAP, Radiance Research Nephelometer, TSI, Inc. | (Mogo et al., 2012) |



**Table 3. Comparison of mean concentration of BC with other studies**

| Site | Environment | Period | Inlet | BC (µg/m³) | Instrumentation | References |
|---|---|---|---|---|---|---|
| Cape D'Aguilar, Hong Kong | Rural, coastal | Feb. 2012-Feb. 2015 | $PM_{2.5}$ | 1.4 | AE31, Magee Scientific | This work |
| Cape D'Aguilar, Hong Kong | Rural, coastal | Jun. 2004-May. 2005 | $PM_{2.5}$ | 2.4 | AE42, Magee Scientific | (Cheng et al., 2006) |
| Yongxing Island, China | Oceanic rural, (South China Sea) | May–Jun 2008 | $PM_{2.5}$ | 0.54 | Aethalometer, Magee Scientific | (Yu et al., 2013) |
| Maofengshan, China | Rural, PRD | May–Jun 2008 | $PM_{10}$ | 2.62 | Aethalometer, Magee Scientific | (Yu et al., 2013) |
| Toulon, France | Semi-urban, coastal | Jun. 2005-Oct. 2006 | $PM_{2.5}$ | 0.95 (winter) 0.45 (summer) | AE31, Magee Scientific | (Saha and Despiau, 2009) |
| Hyytiälä, Finland | Boreal forest. | Dec. 2004-Dec. 2008 | $PM_{2.5}$ | 0.32 | AE31, Magee Scientific | (Hyvarinen et al., 2011) |
| Voerde-Spellen, Germany | Rural | Sep.-Oct. 1997 | $PM_{2.5}$ | 0.8 | AE-10 IM, G1V | (Kuhlbusch et al., 2001) |
| Preila, Lithuania | Rural, coastal | Mar.-Apr. 2002 | $PM_{2.5}$ | 0.84 | AE40, Magee Scientific | (Andriejauskienė, 2008) |



**Table 4. Summary of seasonal average value of target pollutants**

|  | **Winter** | **Spring** | **Summer** | **Autumn** |
|---|---|---|---|---|
| $\sigma_{ap, 550nm}$ **(Mm$^{-1}$)** | 10.9 | 7.5 | 5.5 | 7.4 |
| $\sigma_{sp, 550nm}$ **(Mm$^{-1}$)** | 193.5 | 147.6 | 64.2 | 140.5 |
| **SSA (550nm)** | 0.94 | 0.93 | 0.90 | 0.94 |
| **BC ($\mu$g/m$^3$)** | 2.0 | 1.3 | 0.9 | 1.5 |
| **CO (ppbv)** | 459.0 | 275.3 | 116.6 | 269.6 |
| **AAE** | 1.12 | 0.98 | 0.73 | 0.95 |
| **SAE** | 1.31 | 1.18 | 1.44 | 1.67 |
| **N$_{total}$ (#/cm$^3$)** | 7691 | 8622 | 7003 | 7808 |
| **GMD (nm)** | 71.6 | 68.0 | 57.8 | 69.3 |
| **m$_{0.8}$ ($\mu$g/m$^3$)** | 27.1 | 24.6 | 13.1 | 21.9 |