# Peer review of "Observations of aerosol optical properties at a coastal site in Hong Kong, South China"

_Atmospheric Chemistry and Physics, 2016_

## Referee Comment (RC1) · Anonymous Referee #1 · 30 Nov 2016

General comments: Field observations of aerosol optical properties in different regions are needed due to the spatial and temporal variations in aerosol optical properties which are important to assess the aerosol radiative forcing. In this study, a comprehensive research on aerosol optical properties at a coastal station in Hong Kong based on more than two years' field observation was presented. As similar studies on aerosol optical properties at the same site was conducted a decade ago, this work is necessary and meaningful to reveal the current aerosol optical properties and their variations over the past decade in Hong Kong. Meanwhile, long-term observations of several key aerosol optical properties including AAE, SAE and SSA and studies on the relationships between optical properties and particle size were presented in this study, which were limited in Hong Kong over the past decade. In addition, a range of methods including the ratio of $\Delta BC/\Delta CO$ and $SO2/BC$, LPDM and PSC analysis were employed

to intercept the temporal variations in aerosol optical properties and their quantitative linkage to multi-scale transport. The important influence of ship emissions on aerosol optical properties at Hok Tsui was presented, especially under the southwesterly winds prevailed condition in summer. Overall, this manuscript is well organized and discuss aerosol optical properties and their variation in detail associated with the source analysis. The subject of this study is within the scope of this journal. Several specific comments are listed below. A minor revision is needed before being accepted.

Specific comments: (1) There is a little confusion of the logicality of the sentence 'Correlation analysis showed that the darkest aerosols were smaller in particle size but showed strong scattering wavelength dependencies...' in Lines 18–19 on Page 1. Small particles should have high scattering Angstrom exponent, i.e., strong scattering wavelength dependencies. (2) A PM2.5 cutoff was deployed for the measurement of Aethalometer, thus, the absorption coefficients of PM2.5 were measured. However, the scattering coefficients of TSP were measured by Nephelometer. Which size did the derived SSA represent? PM2.5 or TSP? (3) As stated in Lines 15–17 on Page 8, the SSA at Hok Tsui was slight lower than that observed at a coastal station in Norway in summer (0.91±0.05). However, the average SSA was 0.93 as presented by the authors in Lines 13, higher than the value of 0.91. (4) How the authors concluded that the data clusters with modeled and measured aerosol scattering coefficients fit close to 1:1 is most probably associated with polluted continental air? Have the authors analyzed the retroplume associated with these clusters? (5) Are the absorption and scattering coefficients listed in Table 2 all measured at the same wavelength of 550 nm? Moreover, did the site 'Cape D'Aguilar' in Table 2 represent 'Hok Tsui' station? (6) The diurnal variations of SSA, AAE and SAE are recommended to be presented. Because these parameters represent the aerosol optical properties better which are independent of the absolute aerosol concentrations. (7) The definition of selected episodes in Figure 13, i.e., GH, SP, NC, AGC, should be illustrated in detail in the manuscript. In other words, what criterions to identify these episodes? (8) The measurement of aerosol optical properties was conducted for more than two years. What about their variations

in different years during the observation period?

Technical corrections (1) The full name of 'LPDM' and 'PSC' in Lines 22 on Page 1 should be introduced as they first appeared. (2) Similar to (1), the full name of the corresponding symbol should be introduced as they first appeared. For instant, the full name of $\sigma$ap should be illustrated as aerosol absorption coefficient before it first appear in Line 15 on Page 7. By contrary, an abbreviation should be used throughout the manuscript once it was defined. For example, an abbreviation of SSA is recommended to be used in Line 31 on Page 3 instead of the full name of single scattering albedo, since this abbreviation has been defined above. The authors was recommended to check through the manuscript to avoid such mistakes.

---

## Referee Comment (RC2) · Anonymous Referee #2 · 17 Dec 2016

The paper presents an interesting study of aerosol optical properties over the coastal site of Hong Kong, using three years of data registered with in-situ near instrumentation operated close to the surface. Both the topic and the dataset analyzed make the study worthy to be published in ACP after some revision from the authors.

General comments

Main issues to be solved include the addition of detailed information on quality assurance and associated uncertainties of the different variables analyzed. Furthermore, the authors must carefully revise the number of significant figures used in Tables 1, 2, 3 and 4 and along the text. Table 4 must include more statics information then the simple average, just for giving an idea of the spreading of the data around the mean. . . .

Detailed comments

On Page 1 line 16 the authors write 150.6 Mm-1 for the average light scattering that according to Table 1 presents a STD of 99.4 Mm-1. This must be change both in the text and in the table by 150+-100 Mm-1. This procedure must be applied to variables that also present an excessive of significant figures, like absorption coefficient, SAE, Ntotal, GMD and m0.8. This suggestion is based on the fact that using an excessive number of significant figures for the STD is not appropriate. Furthermore, the experimental errors and their propagation are also against the excessive number of significant figures both for the STD and Averages, that must be expressed with coherent significant figures.

In the abstract, line 21 the authors use deltaBC/delta CO for the ratio BC concentration over CO concentration, but in the text they change the notation in some cases, this must be carefully revised and corrected.

On Page 2, line 22 the authors use for the first time in the paper the acronyms LPDM and PSC, so they must include their respective meanings.

The reference Cazorla et al., 2013 is missing in the reference list.

On page 2 line 23, the comment should be enriched using the reference: "Valenzuela, A., Olmo, F.J., Lyamani, H., Antón, M., Titos, G., Cazorla, A., Alados-Arboledas, L., Aerosol scattering and absorption Angström exponents as indicators of dust and dust-free days over Granada (Spain). Atmospheric Research, 154, pp. 1-13. 2015".

On page 4 line 26 the meaning of the acronym CAB must be detailed.

As explained in the general comments, it is necessary including information on the uncertainties for the different experimental and derived variables analyzed in this study.

The UFP monitor presents some limitations that have been described in the literature, see for example: - Hillemann, L., Zschoppe, A., Caldow, R., Sem, G. J., and Wiedensohler, A. (2014). An Ultrafine Particle Monitor for Size-resolved Number Concentration Measurements in Atmospheric Aerosols. J. Aerosol Sci., 68:14–24. - Gómez-Moreno,

[Figure]

F.J., Alonso, E., Artíñano, B., Juncal-Bello, V., Iglesias-Samitier, S., Piñeiro Iglesias, M., Lopez Mahía, P., Perez, N., Pey, J., Ripoll, A., Alastuey, A., De la Morena, B.A., García, M.I., Rodríguez, S., Sorribas, M., Titos, G., Lyamani, H., Alados-Arboledas, L., Latorre, E., Tritscher, T., Bischof, O., 2015. Intercomparisons of mobility size spectrometers and condensation particle counters in the frame of the Spanish atmospheric observational aerosol network. Aerosol Sci. Technol. 49 (9), 777e785. http://dx.doi.org/10.1080/02786826.2015.1074656. In this sense, some comments and the appropriate references must be included in the instrument section.

In section 2.4 the authors present information on the data processing followed for deriving m0.8 including a value for the particle density that requires justification and a reference.

Section 3.1 and Table 2 must include additional studies developed with similar instrumentation in other urban locations affected by mineral dust transport: Lyamani., F. J. Olmo, and L. Alados-Arboledas. Physical and optical properties of aerosols over an urban location in Spain: seasonal and diurnal variability. Atmospheric Chemistry and Physics., 10, 239-254, 2010.

The discussion on page 7 on BC, including the comments on temporal trends, could be enriched considering the next reference: Lyamani, H., F.J. Olmo, I. Foyo, L. Alados-Arboledas. Black carbon aerosols over an urban area in south-eastern Spain: Changes detected after the 2008 economic crisis. Atmospheric Environment, Volume 45, Issue 35, Pages 6423-6432, 2011

---

## Author Comment (AC1) · 21 Jan 2017

**Wang et al.: Observations of aerosol optical properties at a coastal site in Hong Kong, South China, Atmos. Chem. Phys. Discuss., doi:10.5194/acp-2016-833, 2016**

**Replies to reviewers' comments**

**Overview**

The authors thank the reviewers for constructive comments, they helped improving the paper. We have replied to all questions raised by the reviewer. The major changes to the paper are that the we have

- added discussion on the uncertainties of the UFP
- discussion on the uncertainties of the calculation of the absorption coefficients
- instead of using the symbol $m_{0.8}$ we use the symbol $PM_1$
- added discussion on particle density

Below the responses are written in cursive letters and the changes to the manuscript are highlighted by yellow.

**Detailed replies to Anonymous Referee #1**

General comments: Field observations of aerosol optical properties in different regions are needed due to the spatial and temporal variations in aerosol optical properties which are important to assess the aerosol radiative forcing. In this study, a comprehensive research on aerosol optical properties at a coastal station in Hong Kong based on more than two years' field observation was presented. As similar studies on aerosol optical properties at the same site was conducted a decade ago, this work is necessary and meaningful to reveal the current aerosol optical properties and their variations over the past decade in Hong Kong. Meanwhile, long-term observations of several key aerosol optical properties including AAE, SAE and SSA and studies on the relationships between optical properties and particle size were presented in this study, which were limited in Hong Kong over the past decade. In addition, a range of methods including the ratio of $\Delta BC/\Delta CO$ and SO2/BC, LPDM and PSC analysis were employed to intercept the temporal variations in aerosol optical properties and their quantitative linkage to multi-scale transport. The important influence of ship emissions on aerosol optical properties at Hok Tsui was presented, especially under the southwesterly winds prevailed condition in summer. Overall, this manuscript is well organized and discuss aerosol optical properties and their variation in detail associated with the source analysis. The subject of this study is within the scope of this journal. Several specific comments are listed below. A minor revision is needed before being accepted.

Specific comments:

There is a little confusion of the logicality of the sentence 'Correlation analysis showed that the darkest aerosols were smaller in particle size but showed strong scattering wavelength dependencies. . .' in Lines 18–19 on Page 1. Small particles should have high scattering Angstrom exponent, i.e., strong scattering wavelength dependencies.

*__Response__: Thanks for the comment. We have corrected this expression in the revised manuscript.*

*'Correlation analysis confirmed that the darkest aerosols were smaller in particle size and showed strong scattering wavelength dependencies…'*

A PM2.5 cutoff was deployed for the measurement of Aethalometer, thus, the absorption coefficients of PM2.5 were measured. However, the scattering coefficients of TSP were measured by Nephelometer. Which size did the derived SSA represent? PM2.5 or TSP?

*Response: Thanks for the comment. To answer this question we added the following text to the methodology, in section 2.3:*

> *'The $\sigma_{sp}$ and $\sigma_{ap}$ data were used for calculating single-scattering albedo $SSA = \sigma_{sp}/(\sigma_{sp} + \sigma_{ap})$. The nephelometer took its sample from a Total Suspended Particle inlet (TSP) but the Aethalometer through a PM2.5 inlet so it may seem somewat uncertain which size range the SSA represents. However, BC is the most important light-absorbing constituent in aerosol particles and it is well known that they are in the submicron size range. In larger particles there might be some light-absorbing dust particles but their contribution at this site can be considered to be negligible. Therefore it is reasonable to claim that the absorption coefficients derived from the aethalometer data represent absorption in the full TSP size range even if there was a PM2.5 inlet for the Aethalometer. And since the scattering coefficients were measured after a TSP inlet it is also reasonable to say that the SSA represents that of TSP.'*

As stated in Lines 15–17 on Page 8, the SSA at Hok Tsui was slight lower than that observed at a coastal station in Norway in summer (0.91±0.05). However, the average SSA was 0.93 as presented by the authors in Lines 13, higher than the value of 0.91.

*Response: Thanks. This sentence was not organized clearly. The average SSA measured in Hok Tsui was 0.94 in autumn (higher than 0.83 measured in Xinken during autumn) and 0.90 in summer (slightly lower than 0.91 in Norway in summer). We have revised this sentence in the manuscript.*

> *'The highest $\sigma_{ap}$ and $\sigma_{sp}$ values were observed in winter (10.9 $Mm^{-1}$ and 193.5 $Mm^{-1}$, respectively), which were more than twice that of summer. Similar pattern was observed in a previous study in Hong Kong in 1997-1999 (Man and Shih, 2001). Compared with other rural/background sites, the average $SSA_{550nm}$ at Hok Tsui was $0.94 \pm 0.03$ during autumn which was higher than that measured at Xinken, PRD, China in the same season (0.83±0.05), while this value was $0.90 \pm 0.06$ in summer which was slightly lower than that observed at a coastal station in Norway in summer ($0.91 \pm 0.05$, Mogo et al., 2012).'*

How the authors concluded that the data clusters with modeled and measured aerosol scattering coefficients fit close to 1:1 is most probably associated with polluted continental air? Have the authors analyzed the retroplume associated with these clusters?

*Response: Thanks for the comment. Yes. We selected the data points where the slopes were close to 1:1 and computed the average retroplume of these data points (shown in the following figure) which helped us to make this conclusion.*

> *'After computing the averaged retroplume of these clusters, it was found that the former data cluster is mostly associated with polluted continental air and the latter with stronger winds*

*and sea salt particles (figures were not shown).'*

[Figure]

Are the absorption and scattering coefficients listed in Table 2 all measured at the same wavelength of 550 nm? Moreover, did the site 'Cape D'Aguilar' in Table 2 represent 'Hok Tsui' station?

*Response: Yes. The absorption and scattering coefficients listed in Table 2 were all at the same wavelength of 550 nm. Absorption coefficients at 550nm were calculated from linear interpolation within 520-590 nm. The site name 'Cape D'Aguilar' in Table 2 was the previous name of 'Hok Tsui' station and it was used in some earlier papers. We have added notes of the name in Table 2 and Table 3.*

The diurnal variations of SSA, AAE and SAE are recommended to be presented. Because these parameters represent the aerosol optical properties better which are independent of the absolute aerosol concentrations.

*Response: Thanks. We have added the diurnal variations of SSA, SAE and AAE in Fig. 4 in the manuscript.*

The definition of selected episodes in Figure 13, i.e., GH, SP, NC, AGC, should be illustrated in detail in the manuscript. In other words, what criterions to identify these episodes?

*Response: Thanks. We picked out the most representative episode days for each kind of source types by analyzing retroplume results and variation trend of aerosol optical properties. We have added illustrations according to the suggestions.*

> *The major source regions were Guangdong and Hong Kong (GH), ship emission (SP), North China (NC), and aged continental area (AGC). The selection of the episodes was done by combining the footprints using LPDM and the variation trend of aerosol optical properties and PM$_{2.5}$.*

The measurement of aerosol optical properties was conducted for more than two years. What about their variations in different years during the observation period?

*Response: Thanks. We have calculated the yearly averages during data processing but due to the short time period of data, we cannot make conclusions about inter-annual variations. If we have longer period of data in the future, we will investigate the inter-annual variations of aerosol optical properties.*

Technical corrections
(1) The full name of 'LPDM' and 'PSC' in Lines 22 on Page 1 should be introduced as they first appeared.

*Response: Thanks. We have corrected it in the revised manuscript.*
 *'Multi-year ==backward Lagrangian particle dispersion modeling (LPDM) and potential source contribution (PSC)== analysis revealed…'*

(2) Similar to (1), the full name of the corresponding symbol should be introduced as they first appeared. For instant, the full name of σap should be illustrated as aerosol absorption coefficient before it first appear in Line 15 on Page 7. By contrary, an abbreviation should be used throughout the manuscript once it was defined. For example, an abbreviation of SSA is recommended to be used in Line 31 on Page 3 instead of the full name of single scattering albedo, since this abbreviation has been defined above. The authors was recommended to check through the manuscript to avoid such mistakes.

*Response: Thanks. We have corrected them and we have checked the symbol using of other parameters according to the suggestion in the revised manuscript.*
 *For instance, '…, the light absorption coefficients ==($\sigma_{ap}$)== at all wavelengths were corrected using the method presented by Collaud Coen et al. (2010) where...'*

**Anonymous Referee #2**

The paper presents an interesting study of aerosol optical properties over the coastal site of Hong Kong, using three years of data registered with in-situ near instrumentation operated close to the surface. Both the topic and the dataset analyzed make the study worthy to be published in ACP after some revision from the authors.

General comments

Main issues to be solved include the addition of detailed information on quality assurance and associated uncertainties of the different variables analyzed. Furthermore, the authors must carefully revise the number of significant figures used in Tables 1, 2, 3 and 4 and along the text. Table 4 must include more statics information then the simple average, just for giving an idea of the spreading of the data around the mean.

*Response: Thanks for the valuable comments. We have added more illustrations on quality assurance and revised the number of significant figures in Table 1, 2, 3, and 4 and in the text according to the suggestions in the revised manuscript. We have also revised the Table 4 containing the AVG ± STD and median value of each parameter.*

> *'In this work, without specific notes BC concentrations refer to the aethalometer data measured at $\lambda = 880$ nm. Sample flow on the Aethalometer display was checked once a week to ensure the flow was within 0.2 LPM of previous week and flow calibration was conducted once a month using an independent flowmeter. The inlet cyclone was cleaned every month.'*

> *\* Revision of Table 4 please refer to the manuscript.*

Detailed comments

On Page 1 line 16 the authors write 150.6 Mm-1 for the average light scattering that according to Table 1 presents a STD of 99.4 Mm-1. This must be change both in the text and in the table by 150+-100 Mm-1. This procedure must be applied to variables that also present an excessive of significant figures, like absorption coefficient, SAE, Ntotal, GMD and $m_{0.8}$. This suggestion is based on the fact that using an excessive number of significant figures for the STD is not appropriate. Furthermore, the experimental errors and their propagation are also against the excessive number of significant figures both for the STD and Averages, that must be expressed with coherent significant figures.

*Response: Thanks. We have changed the number of significant figures of variables according to the suggestion in the Tables and in the text.*

> *For instance: 'At 550 nm, the average light scattering $(151 \pm 100 \ Mm^{-1})$ and absorption coefficient $(8.3 \pm 6.1 \ Mm^{-1})$ were lower than most of other rural sites in eastern China'*

In the abstract, line 21 the authors use deltaBC/delta CO for the ratio BC concentration over CO concentration, but in the text they change the notation in some cases, this must be carefully revised and corrected.

*Response: Thanks. $\Delta BC/\Delta CO$ at Hok Tsui station were calculated as total concentration minus regional baseline (mentioned on Page 11). But for Fig. 7 we used BC/CO because this ratio was calculated from the emission inventory, which means it only represents the emission intensity (no*

*region baseline) for each grid cell. In other word we can also call it ΔBC/ΔCO only if the regional background was zero.*

On Page 2, line 22 the authors use for the first time in the paper the acronyms LPDM and PSC, so they must include their respective meanings.

**Response**: *Thanks. We have corrected it in the revised manuscript.*

*'Multi-year* backward Lagrangian particle dispersion modeling (LPDM) and potential source contribution (PSC) *analysis revealed that these particles were mainly from the air masses moved southward over Shenzhen and urban Hong Kong and the...'*

The reference Cazorla et al., 2013 is missing in the reference list.

**Response**: *Thanks. We have added it in the reference list.*

Cazorla, A., Bahadur, R., Suski, K., Cahill, J., Chand, D., Schmid, B., Ramanathan, V., and Prather, K.: Relating aerosol absorption due to soot, organic carbon, and dust to emission sources determined from in-situ chemical measurements, Atmospheric Chemistry and Physics, 13, 9337-9350, 2013.

On page 2 line 23, the comment should be enriched using the reference: "Valenzuela, A., Olmo, F.J., Lyamani, H., Antón, M., Titos, G., Cazorla, A., Alados-Arboledas, L., Aerosol scattering and absorption Angström exponents as indicators of dust and dust-free days over Granada (Spain). Atmospheric Research, 154, pp. 1-13. 2015".

**Response**: *Thanks. This paper surely contains useful information and we have added this reference according to the suggestion.*

*'The AAE in externally mixed BC-dominated regions have been reported to be around 1 (Anderson et al., 2007; Hegg et al., 2002; Bond and Bergstrom, 2006; Bond et al., 2013), while it is greater than 1 for some organic aerosol from biomass smoke and mineral dust due to their diverse light absorbing abilities at different wavelength ranges (Kirchstetter et al., 2004;Russell et al., 2010;* Valenzuela et al., 2015; *Devi et al, 2016).'*

On page 4 line 26 the meaning of the acronym CAB must be detailed. As explained in the general comments, it is necessary including information on the uncertainties for the different experimental and derived variables analyzed in this study.

**Response**: *Thanks. We have added the name of this station it in the revised manuscript.*

*'...In order to correct the systematic errors of filter-based absorption technique, the light absorption coefficients* ($\sigma_{ap}$) *at all wavelengths were* calculated by *using the method presented by Collaud Coen et al. (2010) where the $C_{ref}$ factor was set to be 4.26 according to the value from* Cabauw (CAB) *station reported in the same paper.* CAB station is located near populated and industrialized areas which was to some extent similar to Hok Tsui station (near most of cities in the Pearl River Delta region). The reported average $C_{ref}$ value at CAB was 4.26 ± 0.11*

*and it varies from 2.60 to 4.75 (Collaud Coen et al., 2010). There was no MAAP or any other reference absorption instrument available so determining $C_{ref}$ at Hok Tsui was not possible and the published mean $C_{ref}$ at CAB station was used. However, to present an upper estimate for $\sigma_{ap}$, the $C_{ref} = 3.51$ calculated for the clean marine site of Mace Head (MHD) (Collaud Coen et al., 2010) was also used and the respective average $\sigma_{ap}$ and SSA are presented in the discussions. Since the $C_{ref}$ is responsible for the largest uncertainty in the calculation of $\sigma_{ap}$ (Collaud Coen et al., 2010) we did not make further uncertainty analyses by using the uncertainties related to the other factors within the algorithm.'*

The UFP monitor presents some limitations that have been described in the literature, see for example: Hillemann, L., Zschoppe, A., Caldow, R., Sem, G. J., and Wieden- sohler, A. (2014). An Ultrafine Particle Monitor for Size-resolved Number Concentration Measurements in Atmospheric Aerosols. J. Aerosol Sci., 68:14–24. Gómez-Moreno, C2F.J., Alonso, E., Artíñano, B., Juncal-Bello, V., Iglesias-Samitier, S., Piñeiro Iglesias, M., Lopez Mahía, P., Perez, N., Pey, J., Ripoll, A., Alastuey, A., De la Morena, B.A., García, M.I., Rodríguez, S., Sorribas, M., Titos, G., Lyamani, H., Alados-Arboledas, L., Latorre, E., Tritscher, T., Bischof, O., 2015. Intercomparisons of mobility size spectrometers and condensation particle counters in the frame of the Spanish atmospheric observational aerosol network. Aerosol Sci. Technol. 49 (9), 777e785. http://dx.doi.org/10.1080/02786826.2015.1074656. In this sense, some comments and the appropriate references must be included in the instrument section.

*Response: Thanks. The reviewer is right and we have added more illustrations about the limitations of UFP with more references.*

*Both the $PM_1$ and the $\sigma_{sp}$ calculated from the number size distributions have uncertainties due to the uncertainties of the UFP monitor. The first is the wide range of particle diameters within the size bins and the use of the geometric mean of the bin limits for the whole bin. This yields the highest uncertainty for the bin that measures particles in the size range 200 – 800 nm as can easily be calculated assuming all particles in that size range were 800 nm instead of the geometric mean 400 nm. This calculation is theoretical in the real atmosphere, however and yields unrealistically high uncertainties and will not be analyzed further. Another source of uncertainty is related to the instrument itself. Hillemann et al. (2014) found that the number concentrations measured by UFPM are typically within a range of ± 20 % from the reference values measured with an SMPS. Also Gómez-Moreno et al. (2015) compared the UFP with an SMPS and found that the size distributions measured by UFPM and SMPS were similar in the sense that the peak concentrations were observed at the same size. In the same study it was also observed that in the size channels corresponding to particle diameters < 100 nm the UFP overestimated the number concentrations and in the two largest channels it underestimated the number concentrations. These are the channels that measure the particles that have the highest mass and that scatter light most efficiently. It may therefore be argued that both the $PM_1$ and the modeled $\sigma_{sp}$ are underestimated.*

In section 2.4 the authors present information on the data processing followed for deriving m0.8 including a value for the particle density that requires justification and a reference.

*Response*: *Thanks. We have add explanations about the reason of using 1.7 g cm$^{-3}$ to calculate m$_{0.8}$. Also, we have changed the abbreviation m$_{0.8}$ into PM$_1$ and reason was explained in the manuscript.*

*'For spherical particles the aerodynamic diameter D$_a$ is calculated from the mobility diameter D$_m$ as $D_a = D_m \sqrt{\rho_p / \rho_0}$ where $\rho_p$ is the density of the particle and $\rho_o$ the density of water. For D$_m$ = 0.8 µm and $\rho_p$ = 1.7 g cm$^{-3}$ this yields D$_a$ = 1.0 µm. In the results, therefore, the mass concentration calculated from the number size distributions was denoted as PM$_1$....*

*...It was mentioned above that the PM$_1$ concentrations were calculated by using the density of 1.7 g cm$^{-3}$ which deserves reasoning. The densities of major inorganic aerosol compounds such as ammonium sulfate and sodium chloride are 1.76 and 2.165 g cm$^{-3}$ (e.g. Tang, 1996). Zhang et al. (2008) estimated that the density of sulfuric acid-coated soot is 1.7 g cm$^{-3}$. Ambient aerosols contain also many unknown compounds such as organics and also some water even after drying to RH < 50 %. Densities of real atmospheric aerosols have been measured in several campaigns. Quinn et al. (2001) determined aerosol densities on a cruise across the Atlantic Ocean. The density of submicron aerosols, averaged from observations at very different regions was 1.73 ± 0.24 g cm$^{-3}$. Pitz et al. (2003) determined the mean apparent particle density of 1.6 ± 0.5 g cm$^{-3}$ for urban aerosol. Saarikoski et al. (2005) found that at a boreal forest site the average density was 1.66 ± 0.13 g cm$^{-3}$. Based on these publications it is reasonable to use the density of 1.7 g cm$^{-3}$ for the estimation of aerosol mass concentration from the number size distributions of particles smaller than 800 nm of mobility diameter. It has to be noted, however that there is uncertanty in it since it was not measured at this site.'*

Section 3.1 and Table 2 must include additional studies developed with similar instrumentation in other urban locations affected by mineral dust transport: Lyamani., F. J. Olmo, and L. Alados-Arboledas. Physical and optical properties of aerosols over an urban location in Spain: seasonal and diurnal variability. Atmospheric Chemistry and Physics., 10, 239-254, 2010.

*Response*: *Thanks. We have included the results from this paper (Please refer to Table 2 in the revised manuscript).*

The discussion on page 7 on BC, including the comments on temporal trends, could be enriched considering the next reference: Lyamani, H., F.J. Olmo, I. Foyo, L. Alados- Arboledas. Black carbon aerosols over an urban area in south-eastern Spain: Changes detected after the 2008 economic crisis. Atmospheric Environment, Volume 45, Issue 35, Pages 6423-6432, 2011

*Response*: *Thanks. The station in the above-mentioned paper is an urban site located in Graanda, Spain. It is so different a location that we consider it not reasonable to include its results in the present paper, however good the paper is.*